# Catabolism of Hydroxyproline in Vertebrates: Physiology, Evolution, Genetic Diseases and New siRNA Approach for Treatment

**DOI:** 10.3390/ijms23021005

**Published:** 2022-01-17

**Authors:** Ruth Belostotsky, Yaacov Frishberg

**Affiliations:** Division of Pediatric Nephrology, Shaare Zedek Medical Center, Jerusalem 9103102, Israel; yaacovf@ekmd.huji.ac.il

**Keywords:** hydroxyproline, primary hyperoxaluria, oxalate, glyoxylate, collagen post-translational modification, prolyl hydroxylase, HIF-1α, small interference RNA, protein compartmentalization

## Abstract

Hydroxyproline is one of the most prevalent amino acids in animal proteins. It is not a genetically encoded amino acid, but, rather, it is produced by the post-translational modification of proline in collagen, and a few other proteins, by prolyl hydroxylase enzymes. Although this post-translational modification occurs in a limited number of proteins, its biological significance cannot be overestimated. Considering that hydroxyproline cannot be re-incorporated into pro-collagen during translation, it should be catabolized following protein degradation. A cascade of reactions leads to production of two deleterious intermediates: glyoxylate and hydrogen peroxide, which need to be immediately converted. As a result, the enzymes involved in hydroxyproline catabolism are located in specific compartments: mitochondria and peroxisomes. The particular distribution of catabolic enzymes in these compartments, in different species, depends on their dietary habits. Disturbances in hydroxyproline catabolism, due to genetic aberrations, may lead to a severe disease (primary hyperoxaluria), which often impairs kidney function. The basis of this condition is accumulation of glyoxylate and its conversion to oxalate. Since calcium oxalate is insoluble, children with this rare inherited disorder suffer from progressive kidney damage. This condition has been nearly incurable until recently, as significant advances in substrate reduction therapy using small interference RNA led to a breakthrough in primary hyperoxaluria type 1 treatment.

## 1. Introduction

Although proline (Pro) and hydroxyproline (Hyp) are often referred to as an amino acids, their correct definitions are imino acids, because they contain a secondary amino group (imine) in the pyrrolidine ring structure. This distinguishes Pro from all other genetically encoded (proteinogenic) open-chain amino acids. Hyp is formed by hydroxylation of Pro residues in the polypeptide chain. The vast majority of these post-translational modifications occur in procollagen, and they are critical for proper assembly and stability of mature collagen. Collagens are the most abundant proteins, accounting for about a third of all proteins in the body, while Hyp accounts for 10% to 15% of all its amino acid residues [1]. Another extremely important role of this modification is discrimination between normoxic and hypoxic conditions by oxygen sensitive hydroxylation of Pro in hypoxia-inducible transcription factor (HIF-1α) and some other related proteins. Hydroxylation of a Pro residue is an irreversible modification. Hyp is not a proteinogenic amino acid, and it cannot be re-incorporated into newly synthesized proteins, therefore, degradation of Hyp-containing proteins replenishes the Hyp pool for further metabolism. This process mainly takes place in the liver and, to some extent, in the kidneys. Considering the huge turnover of collagen, Hyp catabolism is essential for body homeostasis [2]. Hydroxylation of L-proline occurs in either the 3- or 4-position [3,4], and the catabolic pathways of these two derivatives are different. In humans, 4-Hyp, which accounts for about 99% of Hyp, is converted stepwise to glyoxylate and glycolate in the mitochondria, and it finally turns into glycine in peroxisome [5]. Considering that glyoxylate can be oxidized to oxalate (Ox) when accumulated in high concentrations, the metabolism of Hyp can potentially lead to the formation of this extremely harmful compound. Why is Ox production dangerous for the body? This is due to the low solubility of calcium oxalate (CaOx) and the tendency of CaOx crystals to precipitate in the renal tubules following filtration. Massive CaOx depositions often impair kidney function and may lead to systemic oxalosis [6]. In healthy individuals, de novo production of Ox from Hyp occurs infrequently, and Hyp metabolism accounts for approximately one third of endogenous oxalate synthesis, the level of which itself is very low. [7]. Another source that contributes to oxalate synthesis is assimilation of glycolate from nutrition, especially plant food. Apparently, there are additional, yet unknown, precursors of this compound [8,9]. To avoid conversion of glyoxylate to Ox, it must be kept from interacting with lactate dehydrogenase (LDH) in the cytosol. This is achieved by separating the different stages of glyoxylate metabolism in closed cellular compartments (compartmentalization). Glyoxylate formation from 4-Hyp, and that from glycolate, was shown to occur in the mitochondria and peroxisomes, respectively. Thus, depending on which substrate is the main precursor of glyoxylate, alanine: glyoxylate aminotransferase (AGT), an enzyme that catalyzes the final step of 4-Hyp catabolism—the conversion of glyoxylate to glycine has distinct organelle distribution in species with diverse dietary habits [10]. It is localized in the mitochondria of carnivores, where the main contribution comes from dietary collagen degradation and the formation of Hyp. In herbivores, where the major source of glyoxylate comes from dietary glycolate, it is peroxisomal, while in omnivores it has dual localization. Disturbances in the glyoxylate pathway lead to the release or synthesis of glyoxylate into the cytosol and to Ox production catalyzed by LDH. Mutations in three genes encoding enzymes of glyoxylate metabolism lead to primary hyperoxaluria (PH1, 2, or 3), an early onset inherited kidney disease resulting in progressive kidney damage with wide phenotypic variability ranging from isolated kidney stone events to end stage kidney disease in infancy [11,12]. PH1 is the most severe condition, often leading to kidney failure. The only curative treatment for severe PH1 patients is combined liver-kidney transplantation, or, in rare cases, isolated liver transplantation, because oxalate overproduction takes place in the hepatocytes. Substrate reduction therapy (SRT) is considered a promising approach to design PH treatment based on depleting activity of one of the enzymes preceding glyoxylate production [13]. Tremendous success in the development of RNA interference (RNAi) technologies resulted in FDA and EMA approval of lumasiran as the first therapeutic agent for PH1 treatment in November 2020. In the phase 2 and 3 trials, the majority of patients who received lumasiran had normal or near-normal urinary Ox levels after 6 months of treatment [14,15]. SRT based on small molecule inhibitors and additional targets for RNAi, which can expand application to other forms of PH, are currently under development [13].

## 2. Posttranslational Modification of Proline Residue

Hyp is the most abundant post-transnationally modified amino acid residue in animals. The major Hyp containing proteins are collagens, which are the structural proteins of the extracellular matrix. The tertiary structure of various types of fibril-forming collagen is shaped by a triple helix of polypeptide chains containing the repetitive Gly-X-Y motifs, whereas the X and Y positions can contain any amino acid, but mostly are occupied by Pro and 4-Hyp, respectively [4]. Hyp and Pro residues are critical for mechanical stability of collagens. Pro residues account for about 20% of all collagen amino acids, and about half of them are hydroxylated. In addition to collagens, several other proteins are reported to contain Hyp. In at least two of them, (elastin [16] and argonaute-2; Ago2 [17]), Pro hydroxylation increases their stability, while in the case of the HIF-1α, O_2_-dependent hydroxylation of Pro under normoxic conditions predisposes this protein to proteasomal degradation [18,19]. This is how the sensing of molecular oxygen is carried out.

Pro residue is hydroxylated either at the 3- or at the 4-position by two enzyme families that reside in the lumen of the endoplasmic reticulum. These modifications take place prior to the formation of collagen triple helix. Hydroxylation occurs only on helical Pro residues, while Pro residues in the N-terminal and C-terminal telopeptides, which are not part of the Gly-X-Y motif, are not converted to Hyp [20]. This process requires additional substrates—2-oxoglutarate, molecular oxygen, ascorbic acid, and Fe^2+^ (Figure 1A).

The vast majority of collagen modifications are at position 4. This reaction is catalyzed by the enzyme prolyl 4-hydroxylase, composed of two alpha subunits (isoenzymes encoded by genes *P4HA1*, *2*, or *3*) and two beta subunits (encoded by *P4HB*) [20]. While the alpha subunit of prolyl 4-hydroxylase promotes substrate recognition and performs enzymatic activity, its beta subunit is responsible for retention of the tetramer in the endoplasmic reticulum. Unlike other prolyl 4-hydroxylase family proteins, P4HB possesses chaperon activity, thiol oxidoreductase activity, and it is involved in thiol related signaling and Nox NADPH oxidase regulation [21,22]. However, these activities are probably not related to prolyl 4-hydroxylase function.

Hydroxylation at position 3 is rare and is mainly found in type I and type IV collagens, with an occurrence of 2 to 10 per 1000 amino acid residues of collagen. The 3-hydroxylation of Pro in the repeating tripeptide, resulting in Gly-3Hyp-4Hyp, is catalyzed by one of three isoenzymes of prolyl 3-hydroxylase (encoded by *P3H1*, *2*, and *3*) in association with the helper proteins CRTAP and Cyclophilin B (PPIB) [23,24,25]. Oxygen-dependent hydroxylation of HIF-1α is performed by another class of prolyl 4-hydroxylases named EGLN1, 2, and 3 (PHD1, 2, and 3). The oxygen requirement for EGLN-catalyzed hydroxylation and degradation of HIF-1α provides an efficient molecular oxygen sensing mechanism [18,26]. Indeed, EGLN has a relatively high Km for O_2_, which is slightly higher than atmospheric oxygen concentration [19]. Conversely, collagen prolyl hydroxylase P4HA has a low Km for oxygen, and its activity is not affected by moderate levels of hypoxia. Thus, the O_2_ concentration is a rate-limiting factor for EGLN enzymatic activity under physiological conditions, and a small change in the cellular O_2_ concentration is directly translated into the level of prolyl hydroxylation of HIF-1α. EGLN prolyl hydroxylases have additional targets, such as protein kinases AKT1 and DYRK1 [27,28], and Pro hydroxylation can regulate other important proteins related to cellular oxygen sensing and response, including IKK-β, NFKB1 and p53 [29,30].

## 3. Role of Pro Hydroxylation in Proteins

Proline hydroxylation plays diverse roles in modified proteins. These include an increase in mechanical stability, participation in protein–protein interactions, and regulation of cellular hypoxia response.

### 3.1. Hyp in Collagen

Collagens are a group of structural proteins that serve as a molecular scaffold, and these are expressed in connective tissues, providing their mechanical stability and shape. Collagens are deposited in the extracellular matrix, and they interact with cells through several families of receptors, mainly through integrins, dimeric discoidin receptors (DDR), and glycoprotein VI. These interactions lead to the regulation of cell differentiation, proliferation, adhesion, and migration. Currently, 28 vertebrate collagens are known; the expression of some types of collagen is limited to specific tissues, which allows for a variety of biological functions [31]. Collagen’s unique properties—thermal and mechanical stability, and the ability to enter into specific interactions with other proteins—stem from its triple helical organization provided by the repeating Gly-X-Y motifs. Mature collagen, which is a left-hand helix of three polypeptide chains, is called tropocollagen. Several tropocollagen molecules may self-assemble to form fibrils or reticular structures. While the stability of this spiral rod, about 300 nm in length and 1.5 nm in diameter, is provided by the Gly-Pro-4-Hyp repeats, its flexibility is achieved by the presence of defects and breaks in this motif because regions poor in Pro and Hyp form triple helices that are more relaxed [32]. The pyrrolidine rings of Pro and 4-Hyp stabilize this structure, but 4-Hyp provides the additional thermal stability. The melting temperature of tropocollagens is directly proportional to the 4-Hyp content [4]. The existence of 4-Hyp is critical for the stability of the tropocollagen, but this is not fully understood. One of the considerations is that collagen is stabilized mainly by water molecules surrounding collagen that provide a formation of a network of hydrogen bonds between the hydroxyl group of 4-Hyp and main-chain’s oxygens [33]. Another consideration is that 4-Hyp reduces the number of conformations available to the random coil of triple helix [34]. The latest update on the current understanding of this effect is summarized by Taga and colleagues [35].

Hyp plays an important role not only in contributing to the rigidity of collagens, but also in protein–protein interaction. Typical integrin recognition sites often contain 4-Hyp residues [36]. DDR-collagen interaction is also Hyp dependent, and substitution of 4-Hyp in Gly-Pro-4-Hyp repeat was shown to reduce DDR2 binding with collagen [37]. The involvement of 4-Hyp in interactions of collagen with glycoprotein VI is well established [38]. However, recently, it was demonstrated that 3-Hyp plays a major role in interactions of type IV collagen with glycoprotein VI [39].

### 3.2. Role of Hyp in Oxygen-Sensing

The discovery of the ubiquitous oxygen-sensing mechanism in mammalian cells, and the important role of Hyp in this process, culminated in the 2019 Nobel Prize in Medicine and Physiology being awarded to Greg Semenza, Peter Ratcliffe, and William Kaelin [40]. HIF is a transcription factor that activates the expression of hypoxia-induced genes in various cell types. HIF-activated genes are involved in erythropoiesis, angiogenesis, iron homeostasis, glucose uptake, and, finally, in the regulation of cell survival and apoptosis. Kaelin [19] and Ratcliffe [41] both demonstrated that enzymatic hydroxylation of Pro residues in normoxia results in ubiquitination of HIF-1α by the von Hippel–Lindau tumor suppressor protein (VHL), followed by proteasome degradation of the transcription factor. The hydroxylated residues Pro-402 and Pro-564 are involved in VHL binding. In these conditions, the half-life of HIF-1α is extremely short (less than 10 min) [41]. Hypoxia prevents prolyl hydroxylation and degradation of HIF-1α, and it allows binding of the transcription factor to hypoxia response elements in the promoters of hundreds of genes involved in metabolic adaptation to low oxygen. Discovering the mechanism of HIF-1α regulation provides new directions for the development of therapeutic agents that stabilize or inhibit HIF. Recent studies have demonstrated that the HIF prolyl hydroxylase inhibitor daprodustat is an effective therapeutic agent that stimulates erythropoiesis and increases hemoglobin level [42]. Additionally, HIF-1α inhibitors may potentially serve as anti-cancer drugs, since hypoxia is a common feature of solid tumors.

## 4. Collagen Degradation and Further Metabolism of Hyp

Collagen breakdown is carried out by sequential cleavage into dipeptides by matrix metalloproteinases [43]. The terminal and rate limiting stage of collagen degradation is catalyzed by prolidase (PEPD) [44]. This enzyme specifically splits dipeptides with C-terminal Pro or Hyp.

Further Hyp utilization depends on the position of hydroxylation. Trans-3-hydroxy-L-proline (3-Hyp) is thought to be re-converted to L-proline via a two-step pathway: dehydration by trans-3-hydroxy-L-proline dehydratase (L3HYPDH), resulting in the formation of Δ^1^-pyrroline-2-carboxylate [45], followed by Pyr2C-reductase (CRYM) activity [46] (Figure 1B). Conversion of trans-4-hydroxy-L-proline (4-Hyp) to glyoxylate is performed by four sequential reactions (Figure 1C and Figure 2) [47,48,49]. All these reactions are catalyzed by mitochondrial enzymes: proline dehydrogenase 2 (PRODH2 or HYPDH), delta-1-pyrroline-5-carboxylate dehydrogenase (1P5CDH), glutamic-oxaloacetic transaminase 2 (AspAT), and 4-hydroxy-2-oxoglutarate aldolase (HOGA1). However, only two of these, HYPDH and HOGA, are unique for the Hyp pathway, and the remaining two are also involved in Pro metabolism [5]. The final reaction of Hyp metabolism, transamination of glyoxylate to glycine, is catalyzed by alanine-glyoxylate and serine-pyruvate aminotransferase (AGT) encoded by *AGXT* gene (Figure 1C and Figure 2). Hyp is an important source of glycine synthesis in animals [50]. This explains the low levels of urinary Hyp excretion, despite large amount of Hyp generated, due to the turnover of tissue collagen and the breakdown of dietary animal protein [2].

## 5. Evolutionary Aspects of Hyp/Glyoxylate Pathway

Intracellular localization of AGT is species-specific—it is restricted in the mitochondria in carnivores and in the peroxisome in herbivores [10]. For some reason, in the case of peroxisomal AGT, the Hyp pathway is lengthened by two reciprocal reactions: reduction of glyoxylate to glycolate by mitochondrial glyoxylate reductase GRHPR, and oxidation of glycolate to glyoxylate by peroxisomal glycolate oxidase (encoded by *HAO1*) (Figure 2 in blue). How can such “wasteful” redundancy be explained? We proposed that these reciprocal reactions are needed in order to protect glyoxylate from exposure to cytosolic LDH during transportation to the peroxisome [51,52]. If cytosolic glyoxylate is the culprit, and mitochondrial AGT can efficiently convert glyoxylate to glycine, why is herbivorous glycolate not converted to glyoxylate by HAO1 inside the mitochondria? The need for a peroxisomal location of this reaction is because its co-product is a reactive oxygen species, hydrogen peroxide, and peroxisomes are the site where this harmful substance is neutralized [53]. Thus, the localization of HAO1 is invariant, whereas AGT can be located both in the mitochondria and the peroxisome. Indeed, studies conducted in Danpure’s laboratory, and among other groups, showed significant correlation between intracellular AGT distribution and diet, independent of phylogeny of the organisms [10,54,55]. This is particularly impressive when analyzing such taxonomic units as bats, marsupials, and bears, where phylogenetically closely related species have different feeding patterns. Given that each species has only one *AGXT* gene, how does the structure of the gene in different species explain the different localization of the protein? It turns out that most *AGXT* genes have the potential to encode an N-terminal mitochondrial targeting sequence (MTS) of 22 amino acids and atypical peroxisomal targeting tri-peptide (PTS1) at the C-terminal [56]. The presence or absence of MTS within the mRNA open reading frame, and the availability of this N-terminal signal, determines AGT mitochondrial compartmentalization, even in the presence of C-terminal PTS1 [10]. Expression of MTS, which abolishes PTS1 function, was investigated in bats [57], primates [58], and birds [59]. Bats that eat fish, insects, or blood possess intact MTSs. On the contrary, MTS is not included in the open reading frame in unrelated lineages of frugivorous bats [57]. Loss of MTS, among these species, was due to small deletions or insertions in the 5′ region, causing downstream stop codons or mutations in an initiation codon. Mitochondrial targeting could be regulated, not only by the loss of the upstream translation initiation ATG codon, but also by mutations changing the charge of MTS, because the transport of a protein into the mitochondria is directed by positively charged residues. For instance, MTS analysis of *AGXT* genes in primates demonstrated that, in a few branches, 5′ ancestral translation start site (ATG) has been lost, and, in one branch, a stop codon was generated inside the MTS. In other branches, substitutions reduced the number of basic residues or increased the amount of acidic residues in the N-terminal sequence, which is expected to reduce the effectiveness of the MTS [58]. In omnivores, transcription or translation of *AGXT* gene starts from two different in-frame sites [60]. Thus, it has been discovered that, although humans are omnivorous, the exclusively peroxisomal localization of AGT does not match this, and this probably makes the human body more prone to increased hepatic oxalate production. Indeed, derangements in hydroxyproline catabolism, due to genetic aberrations, lead to severe disease, namely primary hyperoxaluria.

## 6. Primary Hyperoxaluria and Hydroxyprolinemia

Primary hyperoxaluria results from excessive Ox production due to the concomitant activity of lactate dehydrogenase (LDH) (Figure 1C and Figure 2). LDH is a cytosolic enzyme whose physiological function is to catalyze the reversible conversion of pyruvate to lactate using NADH. It provides substrates for the Cori cycle, which is a metabolic pathway by which lactate produced by anaerobic glycolysis in muscle is transported to the liver and kidney, and is converted back to glucose [61,62]. To avoid Ox production, glyoxylate synthesized in mitochondria or peroxisome must be further metabolized to glycine and kept from interacting with LDH in the cytosol. Disturbances in the glyoxylate pathway lead to the release or synthesis of glyoxylate in the cytosol, and to Ox production. Humans are able to handle low amounts of Ox, however, its increase (hyperoxaluria) results in CaOx supersaturation in the urine leading to crystal formation. Loss of function of glyoxylate pathway enzymes is known to cause three types of primary hyperoxaluria. PH1 arises from insufficient activity of AGT, which is the enzyme that catalyzes the final step of the pathway that converts glyoxylate to glycine (Figure 2). Similarly, deficiency of PH2-related enzyme GRHPR, which catalyzes the reduction of glyoxylate to glycolate, results in the accumulation of glyoxylate in mitochondria followed by its release into the cytosol. On the contrary, glyoxylate is a product rather than a substrate of the third PH-related enzyme, 4-hydroxy-2-oxoglutarate aldolase (HOGA1) (Figure 2). The mechanism of Ox production associated with HOGA1 deficiency is a matter of debate [52,63,64]. We believe that, in the context of loss of function of HOGA1, its substrate 4-hydroxy-2-oxoglutarate (HOG) is accumulated in the mitochondria and released into the cytosol. Outside the mitochondria, the production of glyoxylate is performed by a cytosolic enzyme with nonspecific 4-hydroxy-2-oxoglutarate activity. This cytosolic aldolase has yet to be identified [52].

It should be mentioned that metabolic defect in the conversion 4-Hyp to Δ1-pyrroline-5-carboxylate resulting in hydroxyprolinemia is considered a benign condition [65]. Hydroxyprolinemia usually results from HYPDH deficiency, however, identifying a patient with persistent hydroxyprolinemia, who failed to have mutations in HYPDH encoding gene, leaves the possibility of additional defects in hydroxyproline catabolism [66]. Deficiency of glycolate oxidase (HAO1), another enzyme in the glyoxylate pathway, results in asymptomatic isolated glycolic aciduria without any apparent related abnormalities [67,68].

## 7. PH Therapy

Until recently, there were no approved treatments for primary hyperoxaluria that could prevent renal failure or significantly reduce other symptoms. The only exception is the use of pyridoxine, the precursor of pyridoxal phosphate, which is an essential cofactor of AGT, for the treatment of PH1 patients with mutations leading to mitochondrial mislocalization of AGT, mostly the p.Gly170Arg mutation [69,70]. Only liver transplantation corrects the underlying metabolic defect in PH1, and, further, in most cases, kidney transplantation is required along with liver transplantation. PH1 patients with compromised renal function or with kidney failure awaiting transplantation require up to six hemodialysis sessions per week [11,12].

New experimental approaches provide promising directions for PH drug development. Ox production represents cell-autonomous metabolic defect where activity-deficient hepatocytes overproduce oxalate, even once a large fraction of the mutant host hepatocytes have been corrected. This poses an additional challenge for the development of PH treatments. Substrate reduction therapy (SRT) is considered a promising approach to design PH treatments based on depleting the activity of one of the enzymes preceding glyoxylate production. Enzyme inactivation/inhibition can be achieved by the use of small molecules, RNA interference, or targeted genomic intervention. A promising candidate for SRT is the top enzyme of Hyp metabolism, HYPDH [5]. In theory, this therapy is suitable for all three types of PH. The same is true for LDH inhibition, which is not SRT but is also based on depleting enzymatic activity. However, the most popular SRT target is HAO1, which is relevant only for the most severe condition, PH1. Considering that isolated glycolic aciduria, resulting from HAO1 deficiency in humans is a benign condition, it is considered a safe therapeutic strategy [67,68]. siRNA is an attractive way to knockdown the expression of target proteins. As a result of the latest breakthrough in PH therapy being achieved through a revolutionary application of this approach [71], we will mainly discuss this methodology, whereas detailed information on the development of other drugs is summarized in recent reviews on this topic [13,72,73]. One of the advantages of applying RNA interference for the treatment of PHs is the possibility of targeted delivery of miRNA to hepatocytes, preventing off-target effects in other tissues. Conjugate-based delivery is designed utilizing N-acetylgalactosamine (GalNAc) ligands that bind to the asialoglycoprotein receptor (ASGPR) specifically expressed by hepatocytes. Subsequently, conjugated synthetic dsRNA, homologous in sequence to the silenced gene, is taken up by endocytosis via ASGPR, and then it escapes from the endosomes into the cytosol. siRNA is incorporated into an RNA-inducing silencing complex (RISC) where the duplex RNA is unwound leaving the anti-sense strand to guide RISC to complementary mRNA [74]. The silencing of the target template is catalyzed by cleaving the mRNA with the enzyme Ago2, which, incidentally, is a rare protein stabilized by prolyl 4-hydroxylation [17]. Lumasiran is a synthetic siRNA aimed to reduce the hepatic production of glycolate oxidase by degrading HAO1 mRNA. In phase 3 trials, lumasiran reduced urinary oxalate excretion in the majority of PH1 patients to normal or near-normal levels after 6 months of treatment [14,15,75]. It is administered subcutaneously in 3 monthly loading doses followed by quarterly doses as of the 4th month. In November 2020, Oxlumo (lumasiran) was approved for the treatment of patients with PH1 in all age groups by the FDA and the EMA.

Nedosiran is another hepatocytes’ targeted siRNA that inhibits LDHA responsible for the final step of oxalate production. The advantage of this approach is that, theoretically, it may be suitable for all types of PH [76]. Pre-clinical experiments in a mouse model of PH1 and PH2 treated with LDHA-specific siRNA resulted in an efficient reduction of CaOx excretion while preventing crystal deposition in mouse kidneys [77]. The first phase of clinical trials of Nedosiran on PH1 and PH2 patients demonstrated a positive effect on Ox production [78,79], however, in phase 2 trials, PH2 and PH3 patients did not achieve the primary endpoint of significant urinary oxalate lowering effect [80,81].

## 8. Conclusions

Hydroxyproline is a non-proteinogenic imino acid that provides important structural and regulatory properties to a limited number of proteins, including collagens, HIF, integrins, and Ago2. In this article, we have described why the post-translational hydroxylation of Pro in various protein substrates is carried out by different types of prolyl hydroxylases, and we have investigated the biological necessity of such modifications. Although only a few types of proteins undergo such modification, Hyp is one of the most common products of protein degradation. Collagen breakdown, and that of other hydroxylated proteins, must be accompanied by the metabolic processing of 4-Hyp, which converts this substance into glycine for further utilization. This pathway, which recruits numerous enzymes, can be complicated by the nonspecific formation of a dangerous compound, namely oxalate. We have discussed, in detail, the primary hyperoxaluria, which is a pathologic consequence of the defects of catabolism of Hyp, and we have described the essence of a new type of treatment for this disease based on innovative RNAi technology.

## Figures and Tables

**Figure 1 ijms-23-01005-f001:**
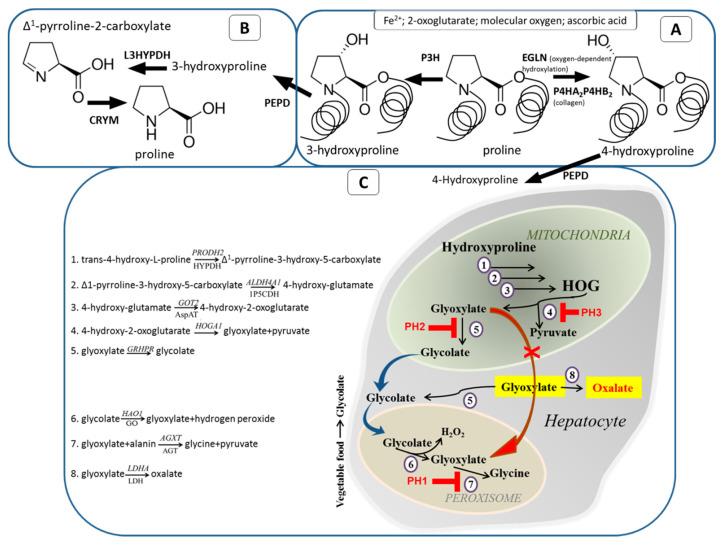
Schematic representation of proline hydroxylation (**A**), catabolism of 3-hydroxyproline (**B**), and 4-hydroxyproline (**C**). Gene designation in italic; protein designation (if different) in regular. PH1, PH2, PH3-types of primary hyperoxaluria.

**Figure 2 ijms-23-01005-f002:**
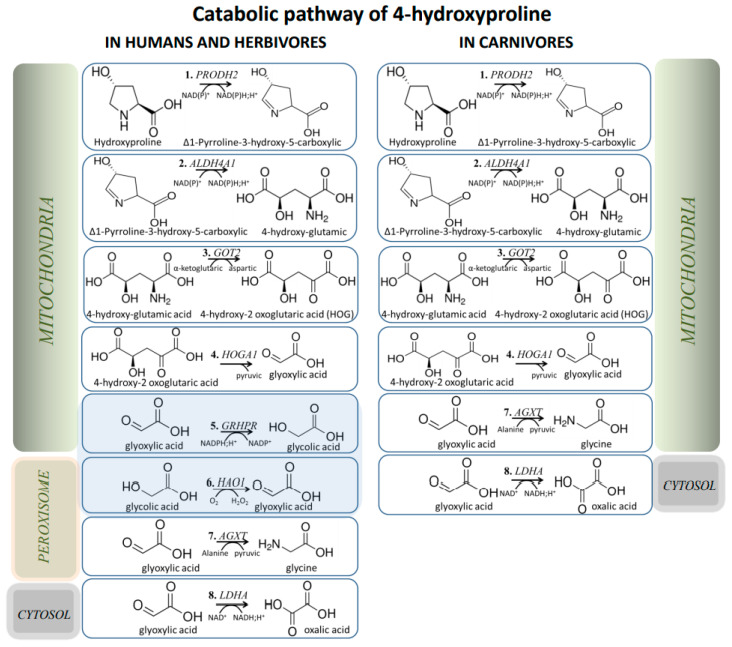
The catabolic pathway of 4-hydroxyproline in various vertebrate species.

## Data Availability

Not applicable.

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
