# Peer review of "Catabolism of Hydroxyproline in Vertebrates: Physiology, Evolution, Genetic Diseases and New siRNA Approach for Treatment"

_ijms, 2022, doi:10.3390/ijms23021005_

Round 1

Reviewer 1 Report

The article is a nice review of a relevant topic in the area of inborn errors of metabolism. It's well written.

Only minor typos should be corrected:

fig 1: alanin --> alanine

line 112: and involved in thiol related signaling --> and is involved in thiol related signaling

line 155: One of considerations is that collagen  --> One of several considerations is that collagen

line 212: the Hyp pathway is lengthens by two reciprocal --> the Hyp pathway is lengthened by two reciprocal

Author Response

We are grateful to the reviewer, appreciate the overall impression and corrected all the typos as suggested.

Reviewer 2 Report

The manuscript 'Catabolism of hydroxyproline in vertebrates: physiology, evolution, genetic diseases and new siRNA approach for treatment’, authored by Ruth Belostotsky and Yaacov Frishberg takes an interesting approach regarding the description of Hyp catabolism, particular in the pathological context. I believe the manuscript could be of potential interest to the readers after some minor modifications:

Major:

  1. A table with the proteins identified as containing 3-/4-Hyp and their corresponding references could be added.
  2. Also the authors could suggest if there are available any strategies to specifically enrich proteins containing Hyp, for large-scale studies, particular in a disease-context.

  3. P2, L45-70: I would suggest to add a figure with detailed chemical structures regarding conversion of 4-HP to glyoxylate/glycolate and further to oxalate with information about the cellular compartments in which these reactions take place. This would help the reader understand much easier the catabolic pathways of 4-HP leading to potential pathologies.

Minor:

  1. Please add a list of abbreviations.
  2. Abstract, p1, l11: Probably should be ‘… cannot be underestimated’ instead of ‘…cannot be overestimated’.

  3. P1, L27: Probably should be ‘…often referred to as amino acids’, instead of ‘… often referred to as an amino acids’.

  4. P5, L212: Probably should be ‘… is lengthen by two …’ instead of ‘is lengthens by two … ’

Author Response

We are very grateful to the reviewer for the thoughtful and creative suggestions. We have addressed all the comments in a point-by-point fashion.

Major:

  1. A table with the proteins identified as containing 3-/4-Hyp and their corresponding references could be added.

Reply: The hydroxyproline proteome performed in recent years has identified a long list of previously unknown proteins that undergo prolyl hydroxylation. For example, Onisco (The Hydroxyproline Proteome of HeLa Cells with Emphasis on the Active Sites of Protein Disulfide Isomerases. J Proteome Res. 2020 Feb 7;19(2):756-768) found 36 unique peptides representing 21 proteins. The authors of another publication found 16 peptides in which proline is presumably hydroxylated by the new PH OGFOD1 (Stoehr A, Kennedy L, Yang Y, Patel S, Lin Y, Linask KL, Fergusson M, Zhu J, Gucek M, Zou J, Murphy E. The ribosomal prolyl-hydroxylase OGFOD1 decreases during cardiac differentiation and modulates translation and splicing. JCI Insight. 2019 May 21;5). Dutta et.al. (Dutta D, Rahman S, Bhattacharje G, Bag S, Sing BC, Chatterjee J, Basak A, Das AK. Label-Free Method Development for Hydroxyproline PTM Mapping in Human Plasma Proteome. Protein J. 2021 Oct;40(5):741-755) detected 11 modified plasma proteins. For a few of these newly discovered Hyp-containing proteins has the biological significance of prolyl hydroxylation been confirmed. However, we still have no data concerning the role of Hyp in most of these proteins, and in many cases even the function of the protein is unknown. We therefore decided to limit the focus of our review to Hyp catabolism and mention only those proteins in which the role of this modification is well established. In our opinion, the discussion of the role of prolyl hydroxylation in recently discovered proteins, as well as the presentation of “strategies for specific enrichment of proteins containing Hyp for large-scale studies” is beyond the scope of this report and merits a separate in-depth review.

  1. Also the authors could suggest if there are available any strategies to specifically enrich proteins containing Hyp, for large-scale studies, particular in a disease-context.

Reply: Please see our reply to the previous item.

  1. P2, L45-70: I would suggest to add a figure with detailed chemical structures regarding conversion of 4-HP to glyoxylate/glycolate and further to oxalate with information about the cellular compartments in which these reactions take place. This would help the reader understand much easier the catabolic pathways of 4-HP leading to potential pathologies.

Reply: Figure 2, illustrating the catabolic pathway of 4-hydroxyproline and localization of its reactions in different compartments, was added, as suggested.

All minor changes were introduced.